# End-Cloud Collaboration Navigation Planning Method for Unmanned Aerial Vehicles Used in Small Areas

**DOI:** 10.3390/s23167129

**Published:** 2023-08-11

**Authors:** Huajie Xiong, Baoguo Yu, Qingwu Yi, Chenglong He

**Affiliations:** State Key Laboratory of Satellite Navigation System and Equipment Technology, Shijiazhuang 050081, China

**Keywords:** navigation planning, unmanned aerial vehicles, end-cloud collaboration, improved particle swarm optimization

## Abstract

Unmanned aerial vehicle (UAV) collaboration has become the main means of indoor and outdoor regional search, railway patrol, and other tasks, and navigation planning is one of the key, albeit difficult, technologies. The purpose of UAV navigation planning is to plan reasonable trajectories for UAVs to avoid obstacles and reach the task area. Essentially, it is a complex optimization problem that requires the use of navigation planning algorithms to search for path-point solutions that meet the requirements under the guide of objective functions and constraints. At present, there are autonomous navigation modes of UAVs relying on airborne sensors and navigation control modes of UAVs relying on ground control stations (GCSs). However, due to the limitation of airborne processor computing power, and background command and control communication delay, a navigation planning method that takes into account accuracy and timeliness is needed. First, the navigation planning architecture of UAVs of end-cloud collaboration was designed. Then, the background cloud navigation planning algorithm of UAVs was designed based on the improved particle swarm optimization (PSO). Next, the navigation control algorithm of the UAV terminals was designed based on the multi-objective hybrid swarm intelligent optimization algorithm. Finally, the computer simulation and actual indoor-environment flight test based on small rotor UAVs were designed and conducted. The results showed that the proposed method is correct and feasible, and can improve the effectiveness and efficiency of navigation planning of UAVs.

## 1. Introduction

Navigation planning is one of the key technologies for UAVs to synergistically perform regional search [1], communication support [2], logistics transportation [3], forest prevention and control [4], and other tasks. Navigation planning refers to the formulation of the optimal flight path from the initial point to the target point via the UAV platform or background assistance after task assignment, meeting the constraints of the UAV performance and environmental conditions, and taking into account the terrain, threats and other factors [5]. The cluster of UAV collaborative navigation planning is different from single-UAV navigation planning, and the factors to be considered are more complex, as they include collision avoidance between UAVs and the arrival of synchronization time in the flight process, making the whole cluster mission benefit the most or the flight cost the least [6]. Conducting research on navigation planning technology can ensure the safe and reliable flight of UAVs in complex environments and achieve mission objectives. In general, the main ways to solve the current UAV navigation planning rely on traditional remote control and ground control stations [7], as well as autonomous sensors and navigation algorithms such as the visual navigation system and inertial navigation system (INS), to achieve simple autonomous control planning [8].

As an important component of the UAV system, the ground control station is the man–machine interface of UAVs [9]. UAVs return information and images to the ground station through the downlink and receive navigation control commands from the ground station through the uplink. One type of control command comprises UAV flight control commands, such as speed, acceleration, heading angle, etc., which can be sent via a remote controller or ground station; another is the flight path target point position information, which is generally sent by the ground station to make the UAV fly to the target position [10]. At present, the common UAV ground station system developed by the mission planner can realize the configuration and simple flight control of the UAV [11]. DJI has developed DJI GS PRO ground station based on iPad, which can realize UAV fixed-path planning and automatic return based on a remote sensing map [12]. In addition to being directly selected by the operator, the location information of the path target point can also be analyzed and calculated by the intelligent algorithm. For example, in [13,14], the deep migration reinforcement learning method and gray wolf algorithm, respectively, were used to achieve the task assignment of multiple UAVs. In [15], the authors used an improved mayfly algorithm to realize UAV path planning. The advantage of background control and navigation planning based on a ground control station is that if accurate environmental situation information can be obtained, the flight path obtained by high background computing power can often be a high-quality solution. However, under the conditions of a complex mission environment and limited communication, it is difficult to ensure the real-time and accurate return of environmental situation information and transmit flight instructions to the front end of the UAV in real time [16].

On the other hand, autonomous navigation based on visual positioning and onboard navigation processing has always been the research focus of UAV navigation control [17]. UAVs realize autonomous positioning through the information fusion of an airborne visual sensor and IMU, and then realize autonomous navigation capabilities such as obstacle-avoidance flight partly through the airborne autonomous path planning algorithm. For example, reference [18] realized obstacle avoidance of a UAV in a dynamic environment based on point-cloud image; reference [19] adopted deep reinforcement learning to realize an end-to-end obstacle-avoidance decision of UAVs. The advantage of navigation decision planning based on airborne autonomous sensors and processors is that the real-time requirement can be satisfied. However, depending on the existing autonomous navigation algorithm in a relatively complex task environment, there are often problems such as insufficient processing capacity and credibility of the autonomous navigation decision of UAVs. In particular, when facing the problem of cluster navigation planning, the key is to design the navigation planning algorithm and technology at a fast speed and with high accuracy [20].

In terms of path planning algorithms, those used at present mainly include three types: traditional algorithms, algorithms based on graph search, and heuristic intelligent search algorithms. Specifically, the authors in [21] proposed a method based on an improved A* algorithm for three-dimensional path planning, and in [22], the coverage path planning problem of autonomous heterogeneous UAVs was studied over a limited number of areas, the region classified into clusters, and approximate optimal point-to-point paths for drones obtained. In [23], the authors used the particle swarm optimization beetle antennae search (PSO–BAS) algorithm to plan the spatial 3D route. The three types of algorithms have their own characteristics. Traditional algorithms can search for the optimal solution, but do not meet real-time requirements due to the high computational intensity. Algorithms based on graph search reduce the computational intensity, but often lack the ability to search for excellent solutions. In terms of balancing the search speed and solution quality, as well as improvement of the algorithm, heuristic intelligent search algorithms have more advantages. The focus of this study is to solve the navigation planning problem of UAVs in complex and communication-limited environments. To this end, an end-cloud collaborative navigation planning algorithm for UAVs was proposed, which combined the background cloud navigation planning algorithm based on an improved PSO and the autonomous navigation control algorithm for UAVs based on a multi-objective hybrid swarm intelligence algorithm. This end-cloud collaborative navigation planning method combines fast real-time path search algorithms on the UAV onboard part and high-quality path planning algorithms on the ground control part, solving the contradiction between the high-quality and real-time path planning that other navigation planning algorithms are difficult to balance. Simulation verification based on software and actual dynamic flight experiment verification based on UAVs and ground station were carried out. The results showed that the algorithm can meet the real-time requirements and improve the accuracy and efficiency of UAV navigation planning. Compared with other navigation planning algorithms, the results show that this algorithm has advantages in search speed and objective function evaluation value.

## 2. System Architecture and Algorithm Design

### 2.1. UAV Navigation Planning Modeling

UAV navigation planning mainly comprises three components: environment modeling, objective function modeling, and navigation planning algorithm design.

Environment modeling includes the total area size and location of the UAV flight tasks, the positions of takeoff start points and mission target points, as well as obstacle and threat information (coordinates, range, altitude, etc.) in the area. Environmental modeling is a prerequisite for UAVs to successfully complete navigation planning. The specific modeling process is shown in Section 3.1.1. The algorithm design part is described in detail in Section 2.2. This section focuses on the modeling and design of the objective function for UAV navigation planning.

The objective function is a key indicator for evaluating the performance of navigation planning, consisting of multiple constraint terms. Constraints usually involve UAVs themself, and the task and environment, which together determine the quality of the navigation planning solution. For the convenience of research, the article does not consider the UAV flight control model, and adds dynamic constraints of UAVs in the objective function model section. The main constraints are as follows:Flight distance

Assuming that the flight speed of the UAV and the average energy consumption remain constant, due to the limited energy capacity of the UAV itself and the limited communication distance with the background, a maximum range must be set to constrain it. On the premise of completing the task, the smaller the range, the better. Assuming that the UAV trajectory consists of N trajectory nodes and N−1 segments, the length of the i  segment trajectory is $\;L_{i}$ and the flight distance constraint of the UAV is: (1)$$L=\sum_{1}^{N-1}L_{i}\leq L_{max}$$
Flight altitude

During flight, the UAV should maintain a low flight altitude to ensure communication with the ground station, but this will increase the probability of colliding with obstacles such as the ground and mountains. Therefore, a minimum value was set to minimize the flight altitude of the UAV, while ensuring that the altitude is not less than this value. The flight altitude constraint is represented as follows:(2)$$H=\sum_{1}^{N}H_{i}\geq H_{min}$$

In the formula, $H_{i}$ is the flight altitude of the UAV at the ith track point and $H_{min}$ is the minimum flight altitude value for UAV.
Distance from the UAV to obstacles

Contrary to flight altitude constraints, in order to ensure effective collision avoidance, the larger the distance between the UAV and the obstacle, the better.
(3)$$\lambda =\sum_{1}^{N}\lambda_{i}\geq \lambda_{min}$$

In the formula, $\lambda_{i}$ is the distance from the UAV to obstacles at the ith track point and $\lambda_{min}$ is the minimum distance value.Distance between UAVs

This parameter is the sum of the distances between UAVs in the cluster during flight. The distance between UAVs should not be small enough to increase the risk of collision, and it should not be large enough to cause communication barriers between UAVs.
(4)$$\rho =\rho_{min}\leq \sum_{1}^{N}\rho_{i}\leq \rho_{max}$$

In the formula, $\rho_{i}\;$ is the average of the distances between the UAV and other UAVs at the ith track point. $\rho_{max}\;$ is the maximum average distance between UAVs that can ensure cluster communication and $\rho_{min}\;$ is the minimum average distance between UAVs to ensure collision avoidance.
Dynamic constraints of UAVs

The flight dynamics constraints of UAVs include the turning radius and climbing angle. When UAVs encounter obstacles or threats, they need to turn or climb to avoid them. When turning, the larger the turning radius while avoiding threats or obstacles, the smoother the trajectory. If the turning radius is too small, it will cause the UAV to lose control and crash. When climbing, in contrast to turning, the smaller the climbing angle, the smoother the trajectory while achieving obstacle avoidance. If the climbing angle is too large, it can also cause the UAV to lose control. The constraints are represented as follows:(5)$$R=\sum_{1}^{N}R_{i}\leq R_{max}$$
(6)$$\theta =\sum_{1}^{N}\theta_{i}\geq \theta_{min}$$

In the formulas, $R_{i}\;$ is the turning radius of the UAV at the ith track point, $R_{max}\;$ is the maximum turning radius value, $\theta_{i}\;$ is the climbing angle of the UAV at the ith track point, and $\theta_{min}\;$ is the minimum climbing angle value.

Based on the above constraint analysis, different constraint terms are selected for combination and coefficient allocation according to different needs. Through single objective or multi-objective optimization, the UAV navigation planning objective function (detailed in Section 2.2.2 and Section 2.2.3, respectively) can be obtained, which is an evaluation function for the quality of path points, used to judge the quality of the solutions searched by the navigation planning algorithm.

### 2.2. UAV Navigation Planning Algorithm Design

#### 2.2.1. End-Cloud Collaboration Navigation Planning Algorithm Architecture

The study designed the architecture of the UAV end-cloud collaborative navigation planning system, as shown in Figure 1. In the background cloud navigation planning part, an improved PSO algorithm is used to plan the approximate track points for the UAVs. In the UAV onboard navigation planning part, an improved multi-objective hybrid swarm intelligence algorithm and B-spline curve are used to be responsible for the local navigation control and path optimization of the UAVs.

The design of the navigation planning algorithm based on the background cloud processor can rely on the high computational power, so the more complex intelligent optimization algorithm can be used. However, the problem of real-time navigation planning due to the limitations of communication links should be taken into consideration, so only the approximate route point planning with lower frequency can be carried out for UAVs. If the communication conditions with front-end UAVs are non-existent, the UAVs can only rely on an airborne intelligent planning algorithm to achieve navigation flight. When the communication link from the front-end to the background cloud is available, the reasonable track points that meet the requirements are planned based on an improved PSO and sent to the front-end UAVs to realize the navigation flight based on the approximate track points.

#### 2.2.2. Background Cloud Navigation Planning Part

Particle swarm optimization (PSO) is an optimization algorithm jointly proposed by Kennedy, Ph.D. of American social psychology, and Eberhart, Ph.D. of electronic engineering in 1995 [24]. In classical PSO, the individual is regarded as a particle without mass and volume, and the target is regarded as the solution of the optimized problem. The movement direction and speed of each particle are affected by their own, as well as group information. The information sharing and cooperation between individuals in the group is used to make the group move toward the direction of the optimal solution, completing the search of the whole particle swarm in the solution space.

The velocity and position update formula of PSO particles are as follows [25]:(7)$$v_{i}^{t}=\omega v_{i}^{t-1}+c_{1}r_{1}\left({pBest}_{i}-x_{i}^{t-1}\right)+c_{2}r_{2}\left(gBest-x_{i}^{t-1}\right)$$
(8)$$x_{i}^{t}=x_{i}^{t-1}+v_{i}^{t-1}$$

$x_{i}^{t}$ represents the position of the ith particle after tth iteration and $v_{i}^{t}$ represents the velocity of the *i*th particle after the *t* iteration. ω is the inertia weight factor, $c_{1}$ and $c_{2}$ are the learning factors, ${pBest}_{i}$ represents the historical optimal position of the ith particle itself, and gBest represents the historical optimal position of the entire population. Usually,$\;N_{s}$ represents the number of particles and $f(x_{i})$ is the evaluation function of the optimized problem. The particle position depends on the value of $f(x_{i})$. Each particle performs the following operations: obtaining the particle position $x_{i}^{t-1}$ after *t* − 1th iteration, the corresponding $f(x_{i}^{t-1})$ can be calculated. According to Formulas (1) and (2), the speed and position are updated and the next iteration is carried out to get $f(x_{i}^{t})$. Then, the above operation is repeated until $f(x_{i})$ meets the requirements or reaches the maximum number of iterations [26].

Figure 2 is a schematic diagram of solving optimization problems with PSO. The search space is two-dimensional. The global optimal solution is at the black spot. The particles are updated from the initial position to the updated position after iteration, where $v_{1}$ is the original velocity of the particle, $v_{2}$ is the velocity caused by pBest, and $v_{3}$ is the velocity caused by gBest. The final velocity of particle v is determined by $v_{1}$, $v_{2},$ and $v_{3}$ together, to make the particles arrive at the updated position from the initial position, and then the speed and position are updated in the same way. The particles will gradually approach the optimal solution position [27].

Formulas (1) and (2) are the speed and position update formula of standard PSO, and the main parameters involved include inertia weight ω, learning factors $c_{1}$ and $c_{2}$, particle population size $N_{s},$ and maximum flight velocity of the particles $v_{max}$. According to traditional parameter settings, it is easy to encounter situations where the algorithm searches too fast and skips over high-quality solutions, or falls into local optima. Faced with the problem of UAV navigation planning, a series of improvements need to be made to these parameters [28].
Inertia weight ω

In PSO, small inertia weight is conducive to the local precise search of the algorithm, while large inertia weight is conducive to the fast global search of the algorithm. Therefore, dynamically changing inertia weight can achieve better optimization results than a fixed value of inertia weight. The improvement methods of this study are as follows:(9)$$\omega =\left\{\begin{array}{c} \omega_{min},\delta \geq \lambda_{1}\\ \omega_{max},\delta \leq \lambda_{2} \end{array}\right.$$

In the above formula, δ is the degree of particle position change after continuous *k* iterations. If the change of particle position after continuous *k* iterations exceeds $\lambda_{1}\;$, then it can be judged that the algorithm is in the fast global search stage. In such case, ω is made into a smaller value, $\omega_{min},$ to slow down the search speed and prevent the algorithm from missing the optimal solution due to the fast search speed. If the change of particle position does not exceed $\lambda_{2}$, then it can be judged that the algorithm has fallen into local optimum, in which case ω is made a larger value, $\omega_{max},$ to improve the search speed of the algorithm and make the algorithm jump out of the local optimum. $k,{\;\lambda }_{1},\;$ and $\lambda_{2}$ are determined by the complexity of the UAV navigation planning problem, and the required search speed and accuracy.Learning factors $\;c_{1}$ and $c_{2}$

Learning factors $c_{1}$ and $c_{2}$ give particles the ability to learn from their own experience and the best ones in the population. A small learning factor makes for a particle search repeatedly outside the target area, and a large learning factor makes the particles fly to or even cross the target area quickly. Similar to inertia weight, ω, the adaptive adjustment strategy is segmented based on particle position changes. When the algorithm is in the global search stage, $c_{1}\;$ is made a larger value, and $c_{max}$, $c_{2}$ a smaller value, $c_{min}$ to prevent the premature convergence of the algorithm. When the algorithm is in the local search stage, $c_{1}$ is made a smaller value, $c_{min},\;{and\;c}_{2}$ a larger value, $c_{max},$ to ensure the diversity of the algorithm during the local search.
(10)$$c_{1}=\left\{\begin{array}{c} c_{max},\delta \geq \lambda_{1}\\ c_{min},\delta \leq \lambda_{2} \end{array}\right.$$
(11)$$c_{2}=\left\{\begin{array}{c} c_{min},\delta \geq \lambda_{1}\\ c_{max},\delta \leq \lambda_{2} \end{array}\right.$$Population size and topological structure second bullet

For PSO, a large population size means that the particles have strong cooperation performance and high global search ability, but the search time of the algorithm is long. However, a small population size will reduce the available global information of the algorithm, which will lead to premature convergence and fall into local optimization. The population topology of PSO represents the way that a single particle connects with other particles, and represents which particles can share information and cooperate with the iteration. Generally, topological structures can be divided into global and local types. Global topology means that each particle can share and exchange information with any other particle, which will speed up the convergence of the algorithm, but it is easy to fall into the local optimum. On the contrary, local topology can only exchange and share information with neighboring particles. This topology will slow the convergence of the algorithm, but it is not easy to fall into local optimum. The two population topologies are shown in Figure 3. The disadvantage of the fixed population size and topology strategy is that it cannot balance the contradiction between a global search and local search, and cannot achieve a good optimization effect in the UAV navigation planning problem [29].

The study proposed a population size and topology improvement strategy that can adaptively change population characteristics based on the current iteration update status of particles, as shown in Figure 4.Maximum particle flying speed, $v_{max}$

The maximum speed of traditional PSO is a fixed value, but it is difficult to consider both algorithm convergence speed and search accuracy at the same time. Therefore, a speed regulation mechanism improvement strategy is proposed as follows:(12)$$v_{max}=\left\{\begin{array}{c} v_{1},\;\delta \geq \lambda_{1}\\ v_{2},\delta \leq \lambda_{2} \end{array}\right.$$

In the above formula, $v_{1}>v_{2}$. The speed regulation mechanism can meet the requirements of both a global fast search and local accurate search, and can flexibly adjust parameters to meet the versatility of the algorithm in navigation planning.

Using the two-dimensional Ackley function as the objective function, a performance comparison test was conducted on the traditional PSO and the improved PSO. Figure 5 displays the comparison test results, where the red dot is the true maximum and the blue dot is the searched maximum. The test results show that the maximum value searched using an improved PSO is more concentrated and distributed around the true value of 0, while the maximum value searched using traditional PSO is more dispersed, indicating that an improved PSO has better optimization effects.

Section 2.1 has already described the constraints of navigation planning. The article combines flight distance, flight altitude, and distance from UAV to obstacles into an objective function, with the distance between UAVs, turning radius, and climbing angle as boundary constraints. The resulting navigation planning evaluation function is shown by the following formula:(13)$$\left\{\begin{array}{c} f=\sum_{i}\left(\varphi_{1}L_{i}-\varphi_{2}\lambda_{i}-\varphi_{3}h_{i}\right),i=1\ldots N\\ \rho_{min}{<\rho }_{i}<\rho_{max},i=1\ldots N\\ \begin{array}{c} R_{i}<R_{max},i=1\ldots N\\ \theta_{i}>\theta_{min},i=1\ldots N \end{array} \end{array}\right.$$

In the above formula, $\varphi_{1}$, $\varphi_{2}$, and $\;\varphi_{3}$ are the weight coefficients. The influence of various constraints on the navigation planning evaluation function is changed by using artificial weighting methods to adjust their sizes. During the iteration process, the point that minimizes the value of f is the best track point.

The background cloud navigation planning algorithm based on an improved PSO is shown in Figure 6.

#### 2.2.3. UAV Onboard Navigation Planning Part

The UAV onboard navigation planning part mainly solves the autonomous local path planning and navigation control problems of UAVs after the background cloud provides them with approximate flight paths. Moreover, UAV onboard processing platforms are often limited by low computational power, and due to the need for fast algorithm convergence speed in the local navigation control of UAVs, the algorithms are more often to fall into local optimization. Although the above series of improvements to PSO can help solve the local optimization problem of navigation planning, making it easier for the algorithm to search for high-quality solutions, it also increases the computational burden and is not suitable for onboard navigation planning parts with high real-time requirements; hence, the algorithm and optimization method that have fast search speed and low computational complexity are necessarily needed. Based on the advantages of PSO, the study used multi-objective optimization and tabu search to solve the UAV onboard navigation planning problem [30].

(1) Improved multi-objective PSO

The navigation planning evaluation function of Formula (7) is to add various constraints and transform it into single-objective optimization. This approach can obtain an optimal solution to a certain extent when the background cloud computer has sufficient resources, but it cannot meet the requirements of a low computational burden and high real-time performance for UAV onboard navigation planning. Therefore, multi-objective optimization can be adopted to solve this problem.

Multi-objective optimization (MOP) problems can generally be defined as the optimization problem of jointly optimizing multiple objective functions under given constraints. The definition of multi-objective optimization can be expressed in the following standard form [31]:(14)$$\left\{\begin{array}{l} y=\min[f_{1}(x),f_{2}(x),\ldots ,f_{m}(x)]\\ s.t.g_{i}(x)\leq 0 \end{array}\right.$$

In the above formula, $y=\left(y_{1},y_{2},\ldots ,y_{m}\right)\;$ represents the target function, $x=(x_{1},x_{2},\ldots ,x_{n})$ represents decision variables to be optimized, and $\left\{g_{1}(x)\leq 0,g_{2}(x)\leq 0,\ldots ,g_{p}(x)\leq 0\right\}$ represents constraints of the decision variable x.

The difference between multi-objective optimization and single-objective optimization is that a multi-objective optimization problem does not require finding a unique optimal solution; therefore, the complexity and computational time of the algorithm will be reduced and it is more suitable for the onboard navigation planning processing of UAVs with high real-time requirements. In the process of solving multi-objective optimization problems, the key is to find some “better solutions” that can satisfy the situation where each objective function achieves a better performance. This set of solutions is called the Pareto optimal solution set. Pareto domination is defined as follows:

If at least one objective function of the feasible solution p is better than that of the feasible solution q, and all objective functions of p are no worse than that of the individual q, that is:(15)$$\left\{\begin{array}{l} \forall i\in \{1,2,\ldots ,m\},\;\;f_{i}(p)\leq f_{i}(q)\\ \exists i\in \{1,2,\ldots ,m\},\;\;f_{i}(p)<f_{i}(q) \end{array}\right.$$

Then, the feasible solution p Pareto dominates the feasible solution q. Pareto dominance in the application of high-dimensional multi-objective optimization problems will cause a significant increase in the number of non-dominated solutions, making it unable to select a better solution from among them, and the algorithm unable to converge. Therefore, the study proposed a new type of dominance-weighted dominance—combined with the traditional Pareto dominance, that can reduce the computational burden of the algorithm and optimize the performance.

The definition of weighted dominance is given here:

For all objective function values for the problem $f_{1}(x),f_{2}(x),\ldots ,f_{k}(x)$, make:(16)$$F_{i}(x_{1},x_{2})=f_{i}(x_{1})-f_{i}(x_{2})+\rho_{i}\sum_{i\neq j}^{1\ldots k}f_{j}(x_{1})-f_{j}(x_{2})$$

In the above formula, $x_{1}\;and{\;x}_{2}$ are both feasible solutions to the problem and $\rho_{i}$ is the weighted dominance factor. If:(17)$$\left\{\begin{array}{c} \forall i,{\;F}_{i}(x_{1},x_{2})\leq 0\\ \exists i,{\;F}_{i}(x_{1},x_{2})<0 \end{array}\right.$$
then $x_{1}$ is weighted domination $x_{2}$. The weighted dominance factor $\rho_{i}$ is related to the weight ratio of the ith objective function. If the weight of the ith objective function is larger, the weighted dominance factor is smaller and vice versa. Compared to Pareto domination, weighted domination is easier to achieve between particles. This makes it easier for the algorithm to find a better solution through weighted dominance, rather than using a large amount of computing resources to search for the Pareto optimal solution.

An important feature of PSO based on multi-objective optimization is to store all non-dominated solutions generated in a set (called external file) after each iteration according to the dominant relationship between the particles. The global optimal particle position gBest that guides the flight of a particle population is selected from the external file. Based on the weighted dominance, this study proposed an improved external file update strategy, as shown in Figure 7.

The global optimal particle location selection strategy is as follows: If there are particles in the Pareto-dominant position added to the external file for consecutive k generations, it is judged that the diversity of the population is qualified, while the convergence is not qualified. The particles Pareto-dominated in the external file are weighed and sorted, and the particle with the highest ranking in the sorting is selected as the global optimal particle. If no Pareto-dominated particles have been added to the external file for consecutive k generations, it is judged that the convergence of the population is qualified, but the diversity is unqualified, and it is possible to fall into a local optimum. In such case, one of the particles in the external file that is weight-dominated, but not Pareto-dominated, is randomly selected as the global optimal particle.

This external file update strategy can greatly reduce the number of particles entering the external file in each iteration, which is conducive to reducing the computational strength of the algorithm, and the adaptive global optimal particle location selection strategy can balance the relationship between diversity and convergence of solutions, better meeting the needs of UAV navigation planning.

The external file solves the problem of selecting the global optimal location of particles, but the selection of the individual optimal location of particles is still different from the single-objective PSO. It is not to compare the magnitude of the evaluation function values between two particles to determine the good and bad, but to compare the weighted dominance relationship between two particles, which can be divided into the following three situations:If the individual optimal particle in the previous iteration weighted dominates the new particle, the individual optimal particle remains unchanged and the optimal position of the particle is not updated;If the individual optimal particle in the previous iteration is weight-dominated by the new particle, the new particle becomes the individual optimal particle, and the historical optimal position of the particle is updated;If the two are not weight-dominated to each other, a 50% probability is used to choose whether to update the optimal position of the particle or not.

The external file update strategy of the multi-objective particle swarm optimization and the selection strategy of particle global optimal and individual optimal position improvement have been completed. Combined with the hybrid swarm intelligence algorithm below, it can further reduce the computational burden of UAV onboard navigation planning and improve the solution efficiency.

(2) UAV onboard hybrid swarm intelligent algorithm and trajectory optimization

Based on the multi-objective PSO, this section introduced the neighborhood subset of the tabu search and the tabu list, further reducing the computational complexity of UAV onboard navigation planning algorithms, while optimizing the trajectory to meet the flight requirements of UAVs.

The tabu search is a neighborhood search algorithm, characterized by the fact that it is not necessary to search all the solutions in the neighborhood structure every time in the iteration, but to randomly select a subset of the neighborhood to search, the spatial range of each iteration search is small, so it can reduce the computational intensity of the algorithm. Another feature of the tabu search is the setting of the tabu list, which is used to store the generated local optimal solution after each iteration. The solutions in the tabu list will not be searchable in subsequent iterations, and reduce the search solution space, while preventing the algorithm from falling into local optimum. After several iterations, if the aspiration criterion is met, in other words, the evaluation value of the tabu object is better than any solution of the neighborhood subset in that iteration, it will be released and can participate in the next iteration search.

The objective terms of the evaluation function for the background cloud navigation planning of UAVs mentioned in the previous section refer to the length of the flight path L, the distance between the obstacle and the target λ, the flight altitude h, the average distance between UAV ρ, turning radius R, and climbing angle θ. In this section, the UAV onboard navigation planning adopts a multi-objective optimization approach, with the first three items as the objective items and the last three item as the constraint conditions. Therefore, the multi-objective optimization modeling of UAV onboard navigation can be represented as:(18)$$\left\{\begin{array}{c} y=\min\left[\sum_{1}^{M}L_{k},\sum_{1}^{M}\lambda_{k},\sum_{1}^{M}h_{k}\right]\\ \rho_{min}{<\rho }_{k}<\rho_{max},\;k=1,2,\ldots ,M\\ R_{k}<R_{max},k=1\ldots M\\ \theta_{k}>\theta_{min},k=1\ldots M \end{array}\right.$$

The flowchart of the UAV onboard hybrid swarm intelligent algorithm is shown in Figure 8.

Usually, after obtaining the approximate path points planned by the background cloud, the UAVs conduct local navigation planning between each path point to form trajectories. However, due to the flight control system limitation of UAVs, the trajectories cannot be directly used by the UAVs. Therefore, it is necessary to optimize the trajectories obtained by the UAV onboard hybrid swarm intelligence algorithm, and generate smooth trajectories.

The study optimized the planned trajectory based on B-spline curves. B-spline curves can be locally reconstructed from the original trajectory, without changing the overall shape of the trajectory. The B-spline curve is determined by the B-spline basis function. Given n+1 track points and a node vector in space $U=\left[u_{0},{\;u}_{1},\ldots ,{\;u}_{n+k+1}\right]$*,* the k-power B-spline curve is represented as:(19)$$P\left(u\right)=\sum_{i=0}^{n}N_{i}^{k}(u)P_{i}$$

In the above formula, $P_{i}$ is the vertex of the feature polygon that forms the B-spline curve and $N_{i}^{k}(u)$ is the B-spline basis function, using the Cox–deBoor recursive formula as [32]:(20)$$\left\{\begin{array}{c} N_{i}^{0}\left(u\right)=\left\{\begin{array}{c} 1,\;u_{i}\leq u\leq u_{i+1}\\ 0,\;otherwise \end{array}\right.\\ N_{i}^{k}\left(u\right)=\frac{u-u_{i}}{u_{i+k}-u_{i}}N_{i}^{k-1}\left(u\right)+\frac{u_{i+k+1}-u}{u_{i+k+1}-u_{i+1}}N_{i+1}^{k+1}(u) \end{array}\right.$$

In this study, the cubic B-spline curve is used to smooth the trajectory, where k = 3, a B-spline curve can be constructed by every four adjacent track sampling points. By using the formula, it can be obtained that for the four adjacent points $P_{0},{\;P}_{1},\;P_{2},{\;P}_{3}$, there is
(21)$$P\left(U\right)=\frac{1}{6}\left[u^{3}\;u^{2}\;u\;1\right]\left[\begin{array}{cc} \begin{array}{cc} -1 & 3\\ 3 & -6 \end{array} & \begin{array}{cc} -3 & 1\\ 3 & 0 \end{array}\\ \begin{array}{cc} -3 & 0\\ 1 & 4 \end{array} & \begin{array}{cc} 3 & 0\\ 1 & 0 \end{array} \end{array}\right]\left[\begin{array}{c} P_{0}\\ P_{1}\\ \begin{array}{c} P_{2}\\ P_{3} \end{array} \end{array}\right]$$

The specific steps for using B-spline curves to smooth the trajectory are shown in Figure 9.

## 3. Experimental Design and Analysis

For the UAV end-cloud collaborative navigation planning system architecture and improved algorithm proposed in this study, this part carried out simulation comparison experiments and analysis, and used the rotor drones and background cloud navigation planning platform to carry out actual flight experiment verification.

### 3.1. Simulation Experiment Verification and Analysis

Based on the MATLAB environment, the scenario of several UAVs avoiding obstacles and reaching the target point position after navigation planning was simulated. Navigation planning experiments based on the end-cloud collaborative improved PSO (E-CPSO) proposed in this paper, with traditional PSO onboard, two types of improved PSO onboard (improved PSO with parameter changes, PSO combined with genetic algorithm), clustering-based algorithm (CA) [33], and ant colony algorithm (ACA) [34] were conducted, and the effectiveness of navigation planning was compared and analyzed.

#### 3.1.1. Environmental Modeling and Algorithm Parameter Setting

The modeling of the environment includes three parts: flight range, threat area, and obstacle area [35]. The flight range is set to the Cartesian coordinate system area. Threat areas generally refer to electromagnetic interference areas and enemy detection areas. A hemispherical model can be used to model the threat area, which is mathematically described as:(22)$$\left\{\begin{array}{l} W_{i}(x,y,z)=\sum_{i}(x-x_{i}{)}^{2}+(y-y_{i}{)}^{2}+z^{2}=r_{i}^{2}\\ z\geq 0 \end{array}\right.$$

In the formula, $\left(x_{i},y_{i},\;0\right)$ represent the center of the threat area, $W_{i}(x,y,z)$ represents the ith threat area, and $r_{i}$ represents the radius of the threat area. The modeling of obstacle areas adopts the mountain peak model, which is mathematically described as:(23)$$z(x,y)=\sum_{i}h_{i}\exp[-(\frac{x-y_{i}}{x_{si}}{)}^{2}-(\frac{y-x_{i}}{y_{si}}{)}^{2}]$$

In the formula, $\left(x_{i},y_{i}\right)$ represents the geographic center coordinates of the mountain peak, (x,y) represents the coordinates of each point in the terrain projected onto the plane, $h_{i}$ represents the height of the ith mountain, $x_{si},y_{si}$ represent the slope vectors of the peaks in the x-axis and y-axis directions, and z(x,y) represents the height of each point in the terrain.

The terrain parameter settings are shown in Table 1.

The terrain environment model is shown in Figure 10. The red part represents the threat area, and the green parts represent the peaks.

The number of UAVs was set to 5, which arrange at (0, 0, 0), and the number of task target points was set to 5, which were spread at (2000, 1200, 900), (2000, 700, 750), (2000, 680, 280), (2000, 1100, 140), (2000, 1400, 500). The results after task allocation are shown in Figure 11. Where * are the target points after task allocation.

As a comparison, the traditional PSO as UAV navigation planning algorithm was unilaterally set according to the most common way, inertia weight ω=0.6, learning factor $c_{1}=c_{2}=2$, number of particle populations $N_{s}=300$, maximum number of iterations T=200, maximum particle flight speed $v_{max}=2$. The parameter settings of the improved PSO with parameter changes (IPSO) was according to the onboard navigation planning section of the article as follows. The parameter of the PSO combined with the genetic algorithm (GAPSO) was set in a traditional way: the crossover probability was 0.3 and the mutation probability was 0.1. Other parameters were set according to traditional PSO. The parameters of clustering-based algorithm (CA) were as follows: the number of clusters k = 8, the selection probability β=0.7, and the population size and evolution times were the same as those of IPSO. The parameters of the ant colony algorithm (ACA) were as follows: the pheromone factor weight  α=1, the heuristic factor weight γ=7, and the pheromone evaporation and enhancement coefficient $\;\mu_{1}=0.3$, $\mu_{2}=1$.

The parameter settings for E-CPSO proposed in this article were as follows. In the dynamic parameter adjustment strategy, considering the actual needs of UAV navigation planning, $\lambda_{1}=0.3,\lambda_{2}=0.1,k=5$, the maximum and minimum inertia weight $\omega_{max}=0.9,$ $\omega_{min}=0.4$, the maximum and minimum learning factors $c_{max}=3.5$, $c_{min}$ = 0.5, the initial population size $N_{s}=300$, and the maximum particle flight speed $v_{1}=3$, $v_{2}=1.5$. The maximum and minimum average distance constraint between UAVs $\rho_{max}=300,\rho_{min}=50,$ the weight coefficients in single-objective optimization functions $\varphi_{1}{=0.2,\varphi }_{2}=0.3,\varphi_{3}=0.1$, the initial external file size in multi-objective optimization $M_{s}=50$, the weighted dominance coefficient $\rho_{1}=1-\varphi_{1}{=0.8,{\;\rho }_{2}=1-\varphi }_{2}=0.7,{{\;\rho }_{3}=1-\varphi }_{3}=0.9$.

#### 3.1.2. Experimental Steps and Result Analysis

The testing steps are as follows:Step1: Start the simulation software and select the UAV end-cloud collaborative navigation planning algorithm;Step2: Set environmental conditions and task target points and allocate them;Step3: Set the parameters of the E-CPSO navigation planning algorithm;Step4: Start UAV background cloud and onboard navigation planning simulation, obtain the approximate trajectory and precise trajectory of UAVs;Step5: Set the parameters of traditional PSO, IPSO, GAPSO, CA and ACA, and start the UAV navigation planning simulation;Step6: Observe and record the navigation planning results of the algorithms;Step7: Draw graphs of the number of iterations and evaluation function separately and conduct comparative analysis;Step8: End the experiment.

The results of UAV navigation planning based on the algorithms are shown in Figure 12. The black dots and connecting lines represent the trajectory results of the navigation plan. From the figures, it can be intuitively seen that the navigation planning result based on E-CPSO was smoother compared to the others, and the track length of each UAV was shorter.

In order to evaluate the performance of the E-CPSO, the relationship between the number of iterations and the objective function value of the four algorithms were compared as shown in the following figure. As the algorithm proposed in this study involves multi-objective optimization, its objective function value was directly calculated from the evaluation value of the track point position. The objective function values of ACA and CA were also calculated based on each track point position.

From Figure 13a, it can be seen that under the same environmental conditions, number of UAVs and location of task target points, the objective function of UAV traditional PSO, IPSO, and GAPSO tended to flatten out at around 80, 90, and 100 iterations. The objective function of the E-CPSO began to flatten after about 50 iterations, indicating that the convergence speed of the latter was faster than that of the formers. Overall, the objective function value of the E-CPSO was smaller and separately decreased by approximately 42%, 25%, and 13% compared to the traditional PSO, IPSO, and GAPSO when convergence was completed, which indicated that the algorithm proposed in this article had better performance in UAV navigation planning. Figure 13b shows that compared with CA and ACA, E-CPSO still had advantages in the rate of convergence and objective function value when convergence was completed.

The terrain parameters and the number of UAVs were changed, and the performance of the algorithms tested separately. The results are shown in Table 2. From the data in the table, it can be seen that the E-CPSO can complete navigation planning of UAVs in different terrain environments, achieved better target evaluation values for track points, and got a lower average number of iterations compared to traditional PSO, IPSO, GAPSO, CA, and ACA. When the number of UAVs increased to 10, the traditional PSO and GAPSO were no longer able to complete navigation planning in complex environments, while the E-CPSO was still valid, though increasing the number of iterations. The results showed that the effectiveness, timeliness, and adaptability of the algorithm proposed in this article are superior than that of the other algorithms.

### 3.2. Verification of Actual Flight Experiments of UAVs

The actual flight tests by using drones and a ground control station computer loaded with the UAV end-cloud collaborative navigation planning algorithm were conducted.

The UAVs used in the experiment were small quadcopter drones, equipped with visual sensors and inertial sensors to achieve environmental perception and self-positioning, along with equipped with communication equipment to receive control commands and transmit image information to the ground station. The patrol speed was set to 3 m/s, the flight speed during obstacle avoidance was 1 m/s, and the flight task time did not exceed half an hour. Figure 14 shows the equipment used in the flight testing.

The flight test site was selected as an indoor environment of 20 m × 15 m × 4 m, which included two simulated obstacles. The drones flew from the starting point to the target point according to the calculated trajectory of the algorithm. Figure 15 shows the indoor drone flight testing site.

The specific steps of the experiment are as follows:Step1: Start drones and ground control station loaded with the UAV end-cloud collaborative navigation planning algorithms;Step2: The ground control station calculates the track point position and sends it to the drones based on prior environmental information;Step3: The drones fly toward the target trajectory point position, continuously calculating and optimizing their own trajectory during the flight process, and transmit the trajectory and observed environmental information back to the ground station;Step4: The ground station further calculates the trajectory points based on the environmental information transmitted by the drone and sends them to the drone;Step5: Repeat step 3 and 4 until the drones complete obstacle avoidance and reach the final target point, and display the dynamic trajectory of the drones.

The test trajectory results of navigation planning of the two drones were, respectively, displayed in rviz as shown in Figure 16. From the figure, it can be seen that the two drones loaded with the E-CPSO started from the starting point, flew along the trajectory planned by the algorithm, avoided obstacles and reached the task area, and ultimately reached the target point. The experimental results showed that the algorithm is correct and feasible, and can meet the needs for drone navigation planning.

## 4. Discussion

The study proposed an algorithm of UAV end-cloud collaborative navigation planning, combining the onboard navigation planning part of UAVs with the background control station part. In addition, an improved PSO for the navigation control onboard and a multi-objective hybrid swarm intelligent navigation planning algorithm for the navigation control of background were, respectively, designed.

For the algorithm proposed in the article, simulation testing and actual flight experiment verification of UAVs were conducted. The simulation results showed that the algorithm converged with lower objective function values and fewer iterations compared to traditional PSO, IPSO, GAPSO, CA, and ACA navigation planning methods, indicating that the optimization ability and timeliness of the algorithm were significantly improved, and the actual flight test results of UAVs showed that the algorithm can meet the needs of UAV navigation planning in mission and obstacle environments.

There are still some shortcomings in the application of the algorithm in navigation planning under the limited communication resources of UAVs. Follow-up studies will focus on the actual communication situation and other constraints under practical application conditions to improve the algorithm, conducting actual flight tests with more drones in multi-task scenarios.

## Figures and Tables

**Figure 1 sensors-23-07129-f001:**
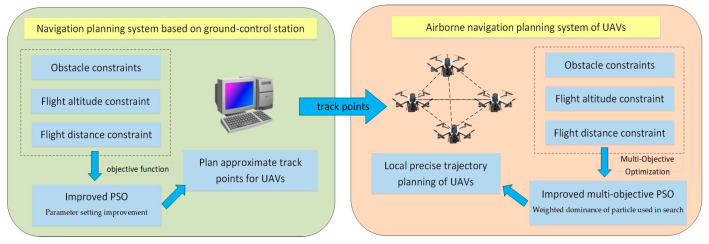
End-cloud collaborative navigation planning system architecture.

**Figure 2 sensors-23-07129-f002:**
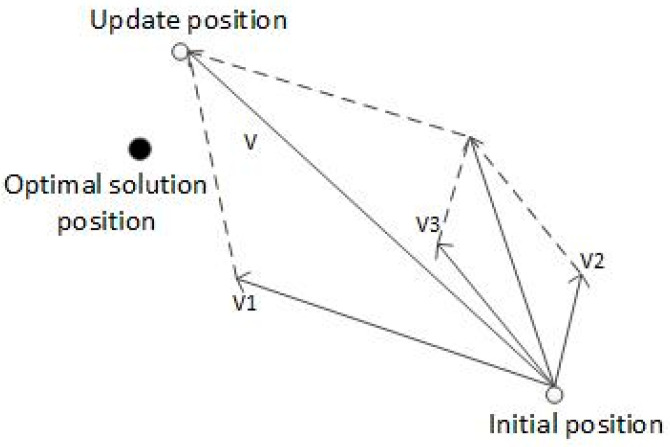
Schematic diagram of PSO iterative updating.

**Figure 3 sensors-23-07129-f003:**
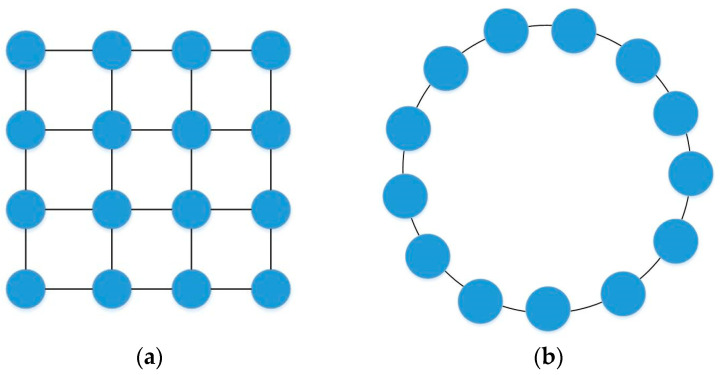
Population topology of PSO: (**a**) global topology; (**b**) local topology.

**Figure 4 sensors-23-07129-f004:**
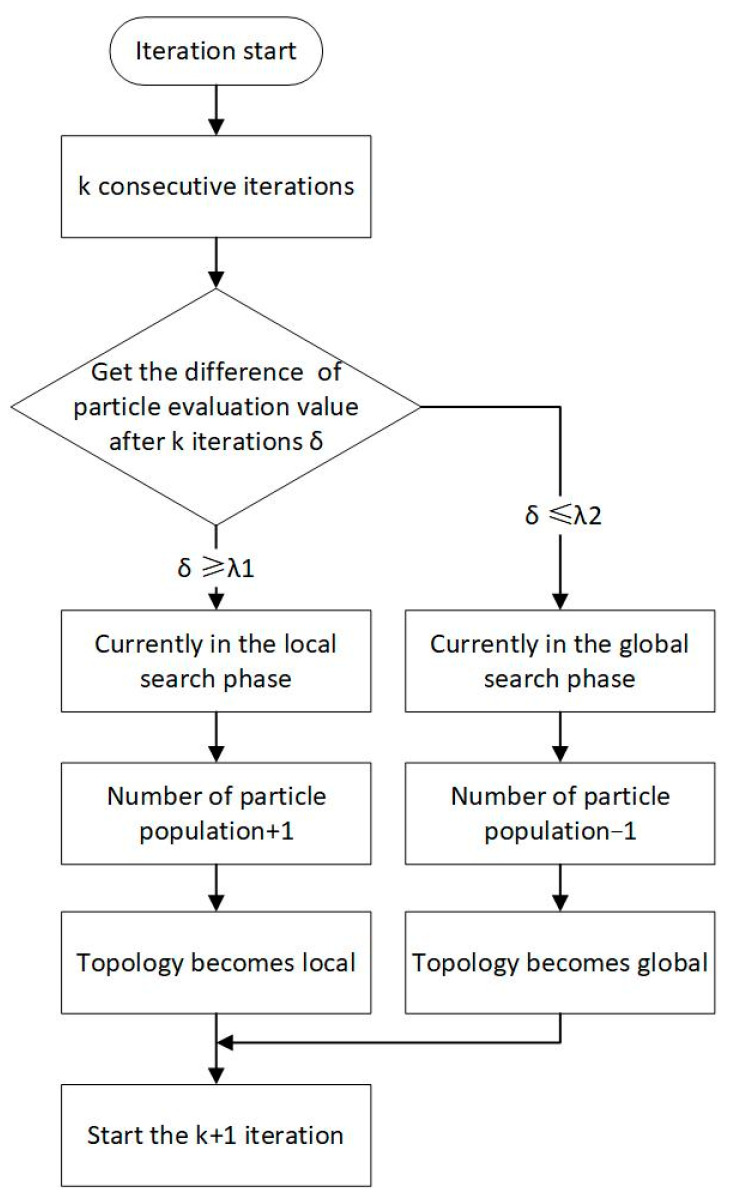
Population size and topology improvement strategies.

**Figure 5 sensors-23-07129-f005:**
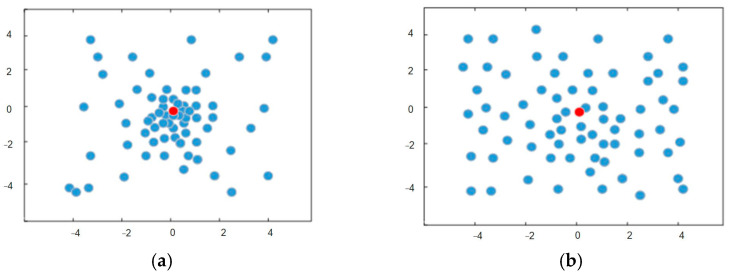
Comparison of algorithm optimization effects: (**a**) by an improved PSO; (**b**) by traditional PSO.

**Figure 6 sensors-23-07129-f006:**
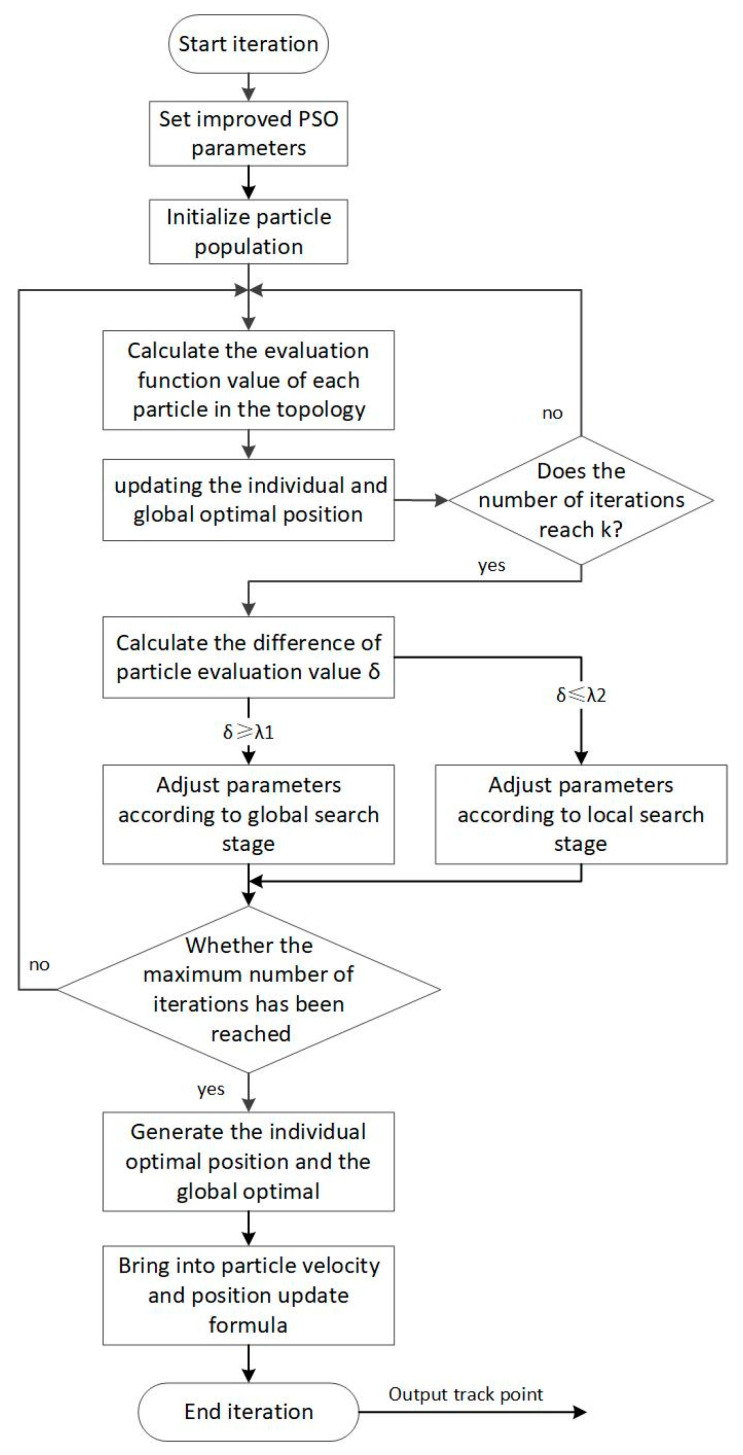
Background cloud navigation planning algorithm based on an improved PSO.

**Figure 7 sensors-23-07129-f007:**
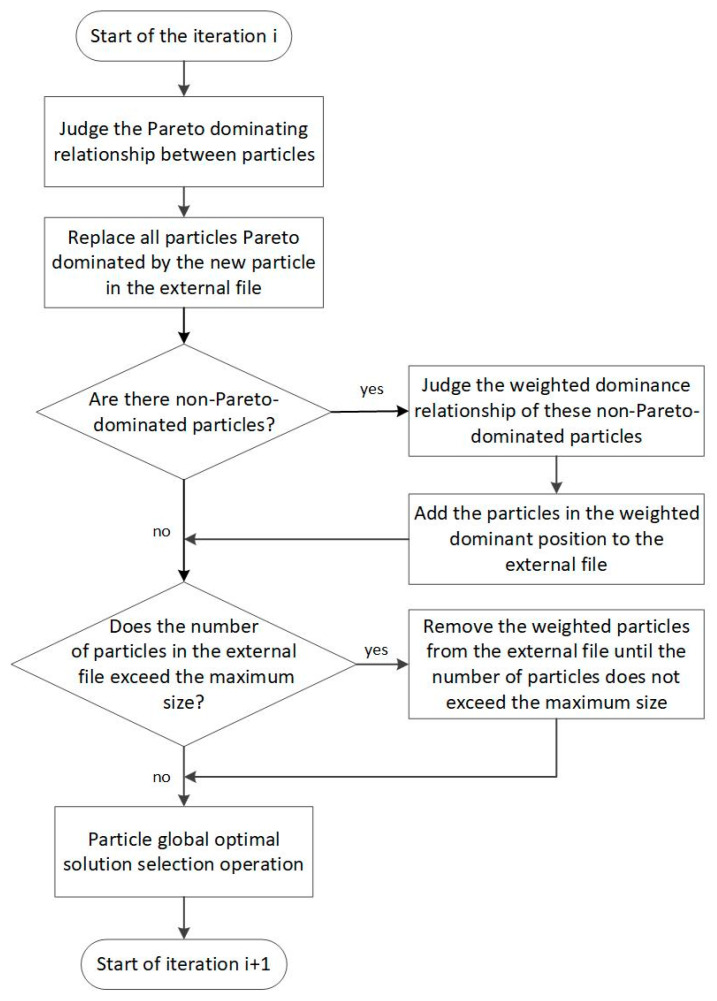
Improved external file update strategy.

**Figure 8 sensors-23-07129-f008:**
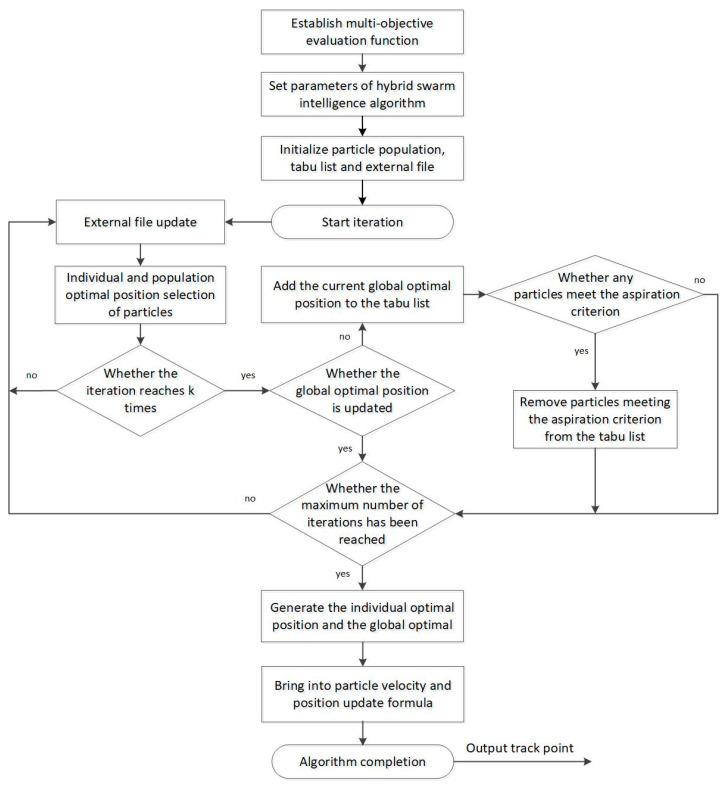
The flowchart of the UAV onboard hybrid swarm intelligent algorithm.

**Figure 9 sensors-23-07129-f009:**
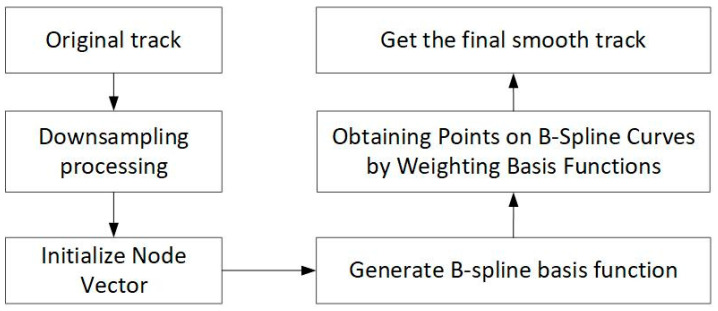
The specific steps for using B-spline curves to smooth the trajectory.

**Figure 10 sensors-23-07129-f010:**
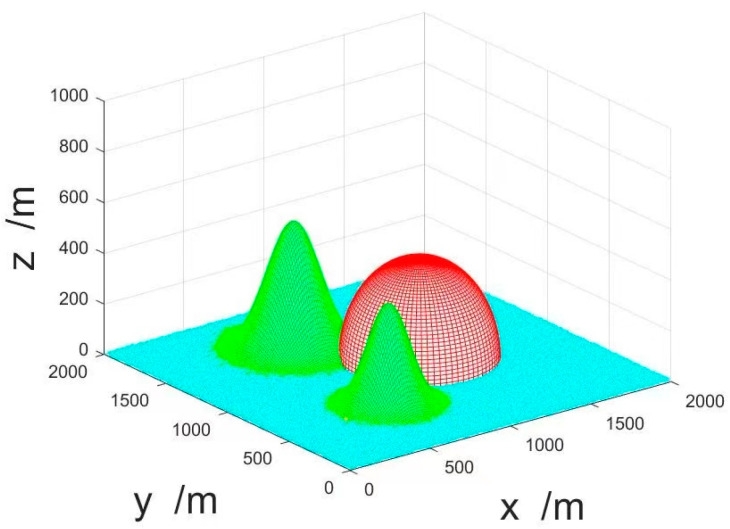
The terrain environment model.

**Figure 11 sensors-23-07129-f011:**
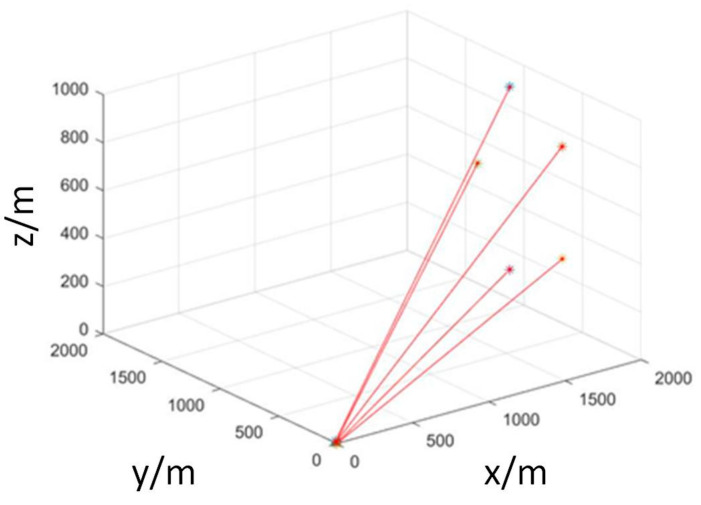
UAV target point task allocation.

**Figure 12 sensors-23-07129-f012:**
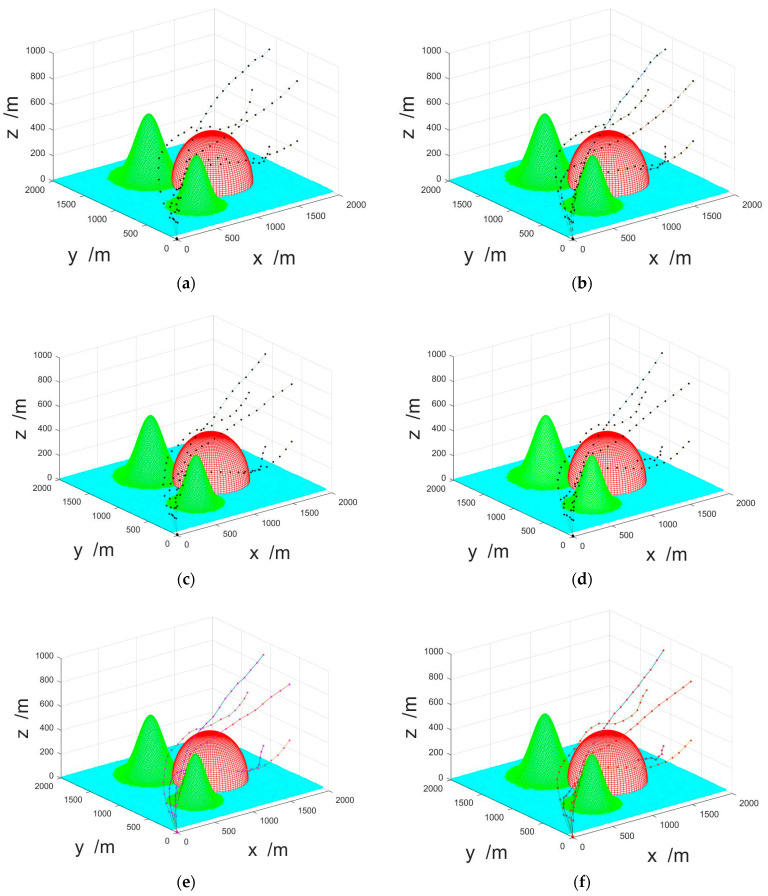
The results of UAV navigation planning based on the four algorithms: (**a**) E-CPSO; (**b**) traditional PSO; (**c**) IPSO; (**d**) GAPSO; (**e**) CA; (**f**) ACA.

**Figure 13 sensors-23-07129-f013:**
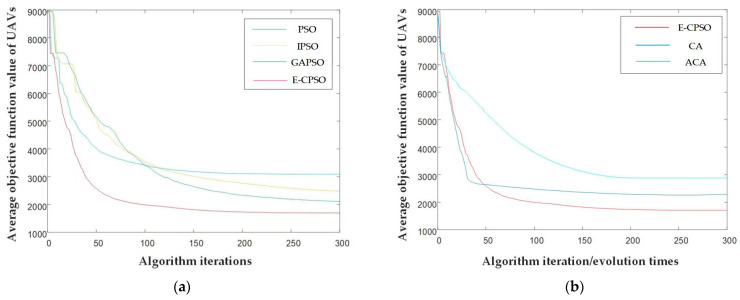
The relationship between the number of iterations and the average objective function value of UAVs of the algorithms: (**a**) E-CPSO compared to PSO, IPSO, and GAPSO; (**b**) E-CPSO compared to CA and ACA.

**Figure 14 sensors-23-07129-f014:**
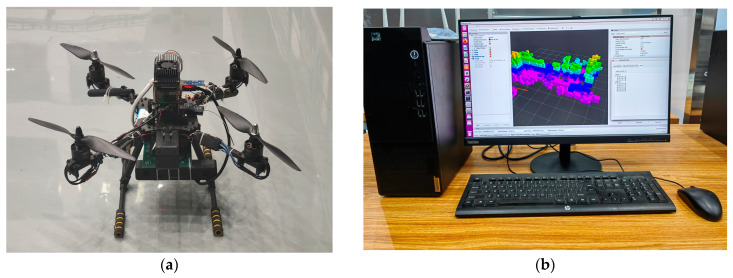
Actual flight testing equipment: (**a**) drones; (**b**) ground control station computer and software.

**Figure 15 sensors-23-07129-f015:**
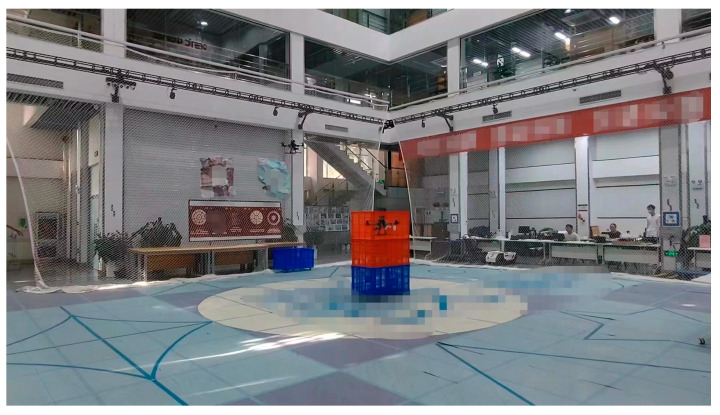
The indoor drone flight testing site.

**Figure 16 sensors-23-07129-f016:**
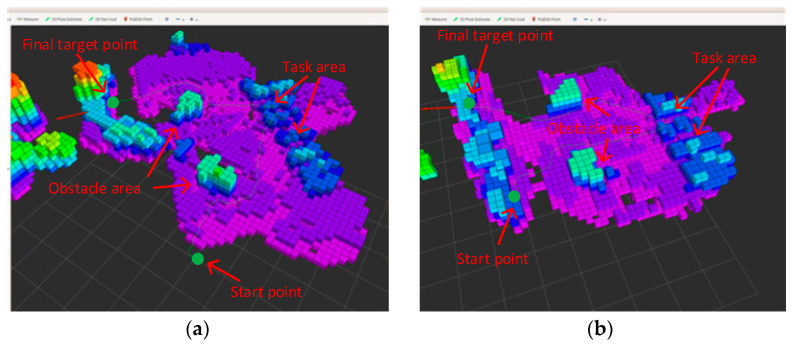
Drone flight trajectory in rviz: (**a**) flight trajectory of drone 1; (**b**) flight trajectory of drone 2.

**Table 1 sensors-23-07129-t001:** Parameters of the terrain.

	Flight Range Parameters (m)	Obstacle Area Parameter OPR ^1^ (m)	Threat Area Parameter TPR ^1^ (m)
Parameters	2000 × 2000 × 1000	(600, 500, 100) (500, 1500, 200)	(1000, 1000, 400)

^1^ where OPR represent the center positions and bottom radius (*x, y, r*) of the obstacle area, and TPR represent the center positions and radius (*x, y, r*) of the threat area.

**Table 2 sensors-23-07129-t002:** Algorithm performance comparison under different conditions.

Terrain Parameter (m)	Algorithm	IBC ^1^ of 5 UAVs	FVC ^1^ of 5 UAVs	IBC ^1^ of 10 UAVs	FVC ^1^ of 10 UAVs
1 threat area, 2 obstacle areas	PSO	82	3128.6	113	6034.5
	E-CPSO	56	1834.4	86	3179.6
	IPSO	88	2465.2	121	4528.3
	GAPSO	96	2104.3	134	4019.1
	CA	72	3104.6	98	6194.4
	ACA	105	2923.5	107	5478.5
2 threat areas, 1 obstacle area	PSO	78	3364.7	121	6823.1
	E-CPSO	52	2023.2	92	4435.3
	IPSO	84	2675.4	125	5287.2
	GAPSO	95	2392.9	138	4963.4
	CA	74	2989.6	111	5892.3
	ACA	96	3045.2	103	6102.2
2 threat areas, 2 obstacle areas	PSO	134	3578.6	--	--
	E-CPSO	112	2149.1	157	4752.8
	IPSO	146	2538.3	172	6272.2
	GAPSO	151	2414.7	--	--
	CA	128	3192.2	169	6490.5
	ACA	123	3234.7	174	6189.8

^1^ where IBC represents iterations at the beginning of convergence and FVC represents objective function value at convergence.

## Data Availability

Not applicable.

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
