# Peer review of "End-Cloud Collaboration Navigation Planning Method for Unmanned Aerial Vehicles Used in Small Areas"

_sensors, 2023, doi:10.3390/s23167129_

Round 1
Reviewer 1 Report
This paper studies the navigation planning problem of UAVs, and presents a novel approach to solve the problem based on end-cloud collaboration systems. From my point of view, the problem studied in this work has good potentials, and also this paper is easy to read in most parts of the text. I suggest a major revision. Main comments are as follows.
1 Since this paper tries to solve the navigation planning problem of UAVs, it is strongly recommended to introduce the purposes or advantages of navigation planning in Section Abstract and Section Introduction.
2 From the view of the reviewer, there is not a part to express the novelty in the whole of the article. The reviewer cannot find whether the idea proposed in this paper is better than those in other papers. Please point out the novelty of the paper in the “Introduction” section.
3 In Section “Introduction”, I feel the current coverage of the state of the art is not satisfactory as the related work section does not cover many contributions. For example, “A clustering-based coverage path planning method for autonomous heterogeneous UAVs” (doi: 10.1109/TITS.2021.3066240), All these works focus on the navigation planning problem of UAVs. It is suggested to cite the above article and analyze the differences.
4 The main content of this paper is Sect. 2, which lists the proposed approach to solve the problem. However, these sections only introduce that how to improve the approach, and does not point out why the authors build the approach in this way. Therefore, Sect. 2 should be reworked carefully. No goal or steps are detailed introduced in this section.
5 It is suggested to split Sect .2 into two section. One section introduces the systems models and the key constraints of the studied problem. The other section detailed introduces the proposed approach.
6 The most advantages of this paper that the performance of the proposed approach is not tested. The authors should compare their approach with some of the existing methods, and show its performance results. In this comment, comparison results of the proposed method and the existing methods are very important. No related method is compared in the current version of paper.
Minor spelling errors should be modified.
Reviewer 2 Report
Schematic diagram of system architecture should be improved, currently it is too general.
Please explain whether it is about planning a static path and optimizing its implementation - e.g., avoiding obstacles indicated by other drones, or dynamic planning - i.e., the path is not known before the flight.
Please explain whether the path planning is related to obstacles of a static nature (e.g., buildings) or of a dynamic nature: gusts of wind, storms.
In abstract one can read: “regional search, railway patrol and other tasks, and navigation planning is one of the key and difficult technologies.” Thus, I can understand that it is about medium to large size drones (very small aircrafts) where accessible energy is limited by liquid fuel volume, and power is mostly consumed by engine not processors of computers. Please give the data of what UAV type are you writing about. On the other side in line 90 I can read “local navigation control and path optimization of the UAVs.”. So I wonder what this paper is about. Local navigation of big drones? What is local navigation? Figure 11 promises 2000km at height of 900km – what a fantastic result -> “deep space UAV “ Please verify me if I’m wrong – UAV in close to International Space Station – please give photo and other data!!!! What a fantastic results.
“DJI has developed DJI 46 GS PRO ground station based on iPad, which can realize UAV fixed path planning and 47 automatic returns based on remote sensing map [12]” - This excerpt suggests that the publication concerns academic models in which drones fly around the iPad. This solution is probably for fun. Somehow, it's hard for me to imagine iPads scattered around the railroad tracks. Maybe somewhere such data is available, and the authors can give it.
Thus, before starting analysis of algorithms please give the energy and power balances of UAV equipment: engines, sensors, computers, transmitters and receivers with respect to planed mission for example as in aviation. Thus, it will be clear what solution is reasonable. On the figure 14 I cannot see the scale of drone. I see a toy drone. Thus, I think this paper is about modeling with verification with toy drone use in closed room. It is not what was suggested in abstract. This decreases the significance of proposed paper. I also suggest to change title to better fit paper contents adding "for small UAVs used in enclosed spaces". Thus, abstract and some sections should be written again.
This publication has potential, but it is too vague in its current form.
Round 2
Reviewer 1 Report
This version is much better than the previous one; however, one of the major problems is the experiment. The authors should compare their approach with more types of the existing methods, and show its performance results.
In the revised manuscript, the authors added some experimental results. However, all comparison baselines are PSO-based approaches. Why did not the authors compare their approach with some other types of solutions. Such as the clustering-based approach presented in ``An Adaptive Clustering-Based Algorithm for Automatic Path Planning of Heterogeneous UAVs'', and the ACS-based approach presented in ``Coverage path planning of heterogeneous unmanned aerial vehicles based on ant colony system''. It is suggested to add these types of solutions into comparison baselines or clearly list the reasons why these approaches are not used.
Some minor errors should be fixed.
Author Response
Please see the attachment. Thank you for your review.

Reviewer 2 Report
The corrected version of the publication is fine. It presents a complete picture of the drone's behavior in a small space when cooperating with a cloud service. I am pleased with such a competent correction of the publication.
Author Response
The simulation experiment section of the paper has been supplemented to improve the comparative analysis of the results. The references have been checked. Thank you for your review.